# A Comprehensive Study of Temperature and Its Effects in SOT-MRAM Devices

**DOI:** 10.3390/mi14081581

**Published:** 2023-08-11

**Authors:** Tomáš Hadámek, Nils Petter Jørstad, Roberto Lacerda de Orio, Wolfgang Goes, Siegfried Selberherr, Viktor Sverdlov

**Affiliations:** 1Christian Doppler Laboratory for Nonvolatile Magnetoresistive Memory and Logic, Institute for Microelectronics, TU Wien, Gußhausstraße 27-29, A-1040 Wien, Austriasverdlov@iue.tuwien.ac.at (V.S.); 2Institute for Microelectronics, TU Wien, Gußhausstraße 27-29, A-1040 Wien, Austria; 3Silvaco Europe Ltd., Cambridge PE27 5JL, UK

**Keywords:** micromagnetics, spintronics, SOT-MRAM, temperature scaling, temperature effects, incubation time

## Abstract

We employ a fully three-dimensional model coupling magnetization, charge, spin, and temperature dynamics to study temperature effects in spin-orbit torque (SOT) magnetoresistive random access memory (MRAM). SOTs are included by considering spin currents generated through the spin Hall effect. We scale the magnetization parameters with the temperature. Numerical experiments show several time scales for temperature dynamics. The relatively slow temperature increase, after a rapid initial temperature rise, introduces an incubation time to the switching. Such a behavior cannot be reproduced with a constant temperature model. Furthermore, the critical SOT switching voltage is significantly reduced by the increased temperature. We demonstrate this phenomenon for switching of field-free SOT-MRAM. In addition, with an external-field-assisted switching, the critical SOT voltage shows a parabolic decrease with respect to the voltage applied across the magnetic tunnel junction (MTJ) of the SOT-MRAM cell, in agreement with recent experimental data.

## 1. Introduction

Magnetoresistive random access memory (MRAM) has recently gained strong attention as a potential replacement for the existing charge-based memories, the static and dynamic random access memories (SRAM and DRAM). Due to the ultra-scaled transistor technology, the leakage currents have rapidly increased together with the static power consumption of the conventional memories. As the information in the MRAM is stored in a relative orientation of two magnetic layers, separated by a thin oxide tunnel barrier, the memory is intrinsically nonvolatile. Hence, the static power consumption is strongly reduced with respect to SRAM and DRAM [1,2]. Moreover, the MRAM is also complementary metal–oxide–semiconductor (CMOS) compatible. In recent years, the two-terminal spin-transfer torque MRAM (STT-MRAM), shown in Figure 1a, has become widely available in the segment of embedded systems. However, due to fairly slow writing times related to the nature of the STT, the writing speed cannot easily reach the sub-nanosecond regimes. Higher currents would partly diminish the problem; however, due to a dielectric breakdown, this would result in faster degradation of the memory cell [3,4].

In order to reach the desired sub-nanosecond switching times and increase the endurance of the MRAM cell, new ways of magnetization manipulation had to be found. In [5], a new method based on the spin Hall effect (SHE) was proposed. Due to the intrinsic characteristics of SOTs, the three-terminal SOT-MRAM, shown in Figure 1b, can operate in the sub-ns region. Moreover, the read and write current paths are separated, and therefore, the endurance is significantly increased. A problem with SOTs for memory application is that, due to the symmetry of the SOTs, the free layer (FL) perpendicular magnetization can only be brought in-plane and another mechanism has to be utilized to complete the switching. These methods include shape symmetry breaking [6], the use of external magnetic fields and built-in magnetic layers [7,8,9], Cr doping to introduce an intrinsic magnetic field [10], two-pulse switching [11], magnetization anisotropy tilt [12], the inclusion of an exchange bias [13], interlayer exchange coupling [14], spin current gradient [15], FL composition gradient [16,17,18], lateral spin-orbit torques [19], competing spin currents [20], out-of-plane spin polarization [21,22,23], ion implantation [24,25], a combination of STT and SOT switching [26,27], or combinations of multiple methods [8,28].

During the writing process of the SOT-MRAM, a strong current passes through the heavy metal (HM). Due to the SHE, a spin accumulation is generated along the sides of the HM. This results in a spin current that is injected into the FL, affecting its magnetization. During the process, however, the current passing through the HM and the FL generates Joule heat and the temperature of the system rises. Consequently, the magnetic properties of the system change, which affects the whole switching process [8]. In [29], the effects of temperature on switching in an SOT system with an exchange bias were studied. Rahaman et al. [30] presented an investigation of the critical switching current with respect to the pulse length duration for an in-plane SOT for different wafer temperatures. In [31], a ferrimagnetic SOT structure is measured and an analysis of the critical switching current is shown. Arpaci et al. [32] then studied the switching of an antiferromagnetic SOT and estimated the device temperature during switching. Even though the mentioned articles discuss temperature and its effects on the switching behavior, none of the articles present a study of the temperature dynamics and its effects on the switching in the ns-regime. Moreover, the listed studies work with μm-sized devices that are several orders bigger than the industry-relevant nm-sized memory cells. In general, the temperature behavior is expected to be different when the size is reduced due to the reduced times scales of the system.

In this work, we focus on the modeling of temperature dynamics and its effects on the switching of the nm-sized SOT-MRAM. We investigate both field-free and field-assisted switching. In Section 2, the used method, implementation, and simulated structures are described. In the first subsection of Section 3, the temperature of an SOT-MRAM cell is analyzed and compared to previous work. The following subsections describe the effects of the increased temperature on the switching.

## 2. Method

In order to model the switching behavior of an SOT-MRAM cell, we fully couple magnetization, charge, spin, and temperature dynamics. We employ the Landau–Lifschitz–Gilbert (LLG) equation to describe the magnetization dynamics.
(1)$$\frac{\partial m}{\partial t}=-\gamma \mu_{0}m\times H_{\mathrm{eff}}+\alpha m\times \frac{\partial m}{\partial t}+\frac{1}{M_{S}}T_{S}$$
where m stands for the normalized magnetization, and γ, $\mu_{0}$, α, and $M_{S}$ are the gyromagnetic ratio, the vacuum permeability, the Gilbert damping, and the saturation magnetization, respectively. $H_{\mathrm{eff}}$ represents the effective field consisting of several components, namely: the demagnetization field $H_{\mathrm{demag}}$, the anisotropy field $H_{\mathrm{aniso}}$, the exchange field $H_{\mathrm{exch}}$, and the external field $H_{\mathrm{ext}}$. $H_{\mathrm{demag}}$ is solved through a hybrid FEM-BEM method [33]. $H_{\mathrm{aniso}}$ is considered to be uniaxial.
(2)$$H_{\mathrm{aniso}}=\frac{2K_{a}}{\mu_{0}M_{S}}\left(n\cdot m\right)n$$
where $K_{a}$ is the anisotropy energy density and n is a unit vector coinciding with the axis of the magnetic tunnel junction (MTJ) cylinder. The exchange field is determined using
(3)$$H_{\mathrm{exch}}=\frac{2A_{\mathrm{exch}}}{\mu_{0}M_{S}}\nabla^{2}m,$$
where $A_{\mathrm{exch}}$ is the exchange stiffness. $H_{\mathrm{ext}}$, when considered, points in the SOT current direction. The spin torque $T_{S}$ represents the torque’s action on the magnetization due to spin-relevant effects and is determined from the spin accumulation S.
(4)$$T_{S}=-\frac{D_{e}}{\lambda_{J}^{2}}m\times S-\frac{D_{e}}{\lambda_{\varphi }^{2}}m\times \left(m\times S\right)$$
where $D_{e}$ stands for the electron diffusion constant, and $\lambda_{J}$ and $\lambda_{\varphi }$ are the spin exchange and dephasing lengths, respectively. Due to the spin dynamics being several orders of magnitude faster than the magnetization dynamics, S can be treated as a static problem [34,35].
(5)$$\frac{\partial S}{\partial t}=0=-\nabla \cdot \bar{J_{S}}-D_{e}\left(\frac{S}{\lambda_{sf}^{2}}+\frac{S\times m}{\lambda_{J}^{2}}+\frac{m\times \left(S\times m\right)}{\lambda_{\varphi }^{2}}\right)$$
(6)$$\bar{J_{S}}=-\frac{\mu_{B}}{e}\beta_{\sigma }m⊗\left(J_{C}-\beta_{D}D_{e}\frac{e}{\mu_{B}}\left[{\left(\nabla S\right)}^{T}m\right]\right)-D_{e}\nabla S-\theta_{SHA}\frac{\mu_{B}}{e}\varepsilon J_{C}$$
where $\lambda_{sf}$ is the spin-flip length, and $J_{C}$ stands for the charge current, while $\mu_{B}$, *e*, $\beta_{\sigma }$, and $\beta_{D}$ are the atomic magnetic moment, the elementary charge, the conductivity spin polarization, and the diffusivity spin polarization, respectively. The last term in (Equation 6) represents the spin Hall effect with spin Hall angle $\theta_{SHA}$. We consider the transverse spin currents to be fully absorbed at the FL/HM interface and we implement a boundary condition based on the real and imaginary part of the mixing conductance $G^{↑↓}$ [35,36].
(7)JS¯·n|N=−2DeσRe(G↑↓)m×(m×S|N)+Im(G↑↓)m×S|N
where σ stands for the electric conductivity and |N indicates the HM side of the interface. Equation (Equation 7) is included as a contribution to the torque in the first layer of elements on the FM side of the interface. The charge current in (Equation 6) is determined from the potential *V* solving (Equation 8) and (Equation 9).
(8)$$-\nabla \cdot \left(\sigma \nabla V\right)=0$$
(9)$$J_{C}=-\sigma \nabla V$$
where σ is assumed to be constant in the ferromagnetic and nonmagnetic layers, while in the tunneling layer, it is assumed to be dependent on the respective angle between the FL and the reference layer (RL) [37,38].

The dynamics of the temperature *T* is modeled using the heat transport equation.
(10)$$c_{V}\rho_{m}\frac{\partial T}{\partial t}-\kappa \Delta T={\dot{q}}_{V}$$
where $c_{V}$, $\rho_{m}$, and *K* are the heat capacity, the material density, and the thermal conductivity, respectively. ${\dot{q}}_{V}$ represents the heat sources, in this case, the Joule heating ${\dot{q}}_{V}=\sigma J_{C}^{2}$. To account for the change in magnetization dynamics, $M_{S}$, $K_{a}$, and $A_{\mathrm{exch}}$ in (Equation 1)–(Equation 3) are made temperature-dependent [8,39]. The $M_{S}$ is scaled according to Bloch’s power law.
(11)$$M_{S}\left(T\right)=M_{S_{0}}m_{S}=M_{S_{0}}\left[1-{\left(\frac{T}{T_{C}}\right)}^{\beta }\right]$$
where $M_{S_{0}}$ is the saturation magnetization at 0 K, and the factor $m_{S}$ represents the scaling with respect to the Curie temperature $T_{C}$ and a temperature power parameter β. The anisotropy constant and the exchange constant also scales with $m_{S}$; however, with additional power coefficients *p* and *q*, and $K_{a_{0}}$ and $A_{{\mathrm{exch}}_{0}}$, the anisotropy and exchange constants at 0 K, respectively.
(12)$$K_{a}\left(T\right)=K_{a_{0}}m_{S}^{p}$$
(13)$$A_{\mathrm{exch}}\left(T\right)=A_{{\mathrm{exch}}_{0}}m_{S}^{q}$$

### 2.1. Implementation

To solve Equations (Equation 1)–(Equation 13), we use the finite element method. The equations are transferred into weak formulations and implemented within our MRAM simulation framework [40]. More details about the weak formulation of the equations can be found in [41].

### 2.2. Simulated Structures

We simulate two different structures. The first one, Structure I, is based on previous simulation work in [8] and is shown in Figure 1a. It consists of a $200\times 230\;{\mathrm{nm}}^{2}$ β−W layer (orange) with a 3.7 nm thickness. On top of the rectangle, an MTJ stack with an 80 nm diameter is placed, consisting of a 1.2 nm thick FeCoB FL (light blue), a 1 nm MgO (red), and a 1 nm FeCoB RL (green). A 12 nm Cu layer is placed on top of the MTJ stack. The whole structure is surrounded by an oxide (half opaque nonhomogeneous gray). Dirichlet conditions are applied (constant 300 K) to the sides of the oxide “box”, which is large enough not to affect the temperature simulation results.

Structure II, depicted in Figure 2b, represents a realistic SOT-MRAM cell [8,42]. The FeCoB FL and the RL (green) are considered to be 1.2 nm and 1.0 nm thick, respectively, the MgO barrier thickness is 1.0 nm, and the MTJ diameter is 40 nm. The β−W heavy metal (depicted in orange) is considered to be 3.7 nm thick and 50 nm wide, the total length is 140 nm. The HM layer is connected through a Cu via to a doped Si substrate (pink). The other end is connected to a long Cu interconnect. The top 50 nm long Cu region is connected to another long Cu interconnect. The whole structure is surrounded by SiO (half opaque nonhomogeneous gray). Both the bottom of the substrate and the ends of the word and read lines are made sufficiently large so as not to have any significant effects on the final result. Dirichlet boundary conditions (constant 300 K) are applied at the ends of the contacts and the bottom of the substrate.

### 2.3. Simulation Parameters

The parameters relevant for the magnetization, charge, and spin dynamics are listed in Table A1 and Table A2, given in Appendix A. We choose the temperature-dependent FeCoB properties so that $M_{S}\left(300\;K\right)=0.81\;{\mathrm{MAm}}^{-1}$, $K_{a}\left(300\;K\right)=539\;{\mathrm{kJm}}^{-3}$ [43], and $A_{\mathrm{exch}}\left(300\;K\right)=20\;{\mathrm{pJm}}^{-1}$ [44]. $T_{C}=750\;K$, β=1.7, p=3, and q=1.7 [8,45]. The temperature-relevant parameters for different materials are listed in Table A3. We consider the parameters to be constant for the simulated temperature range.

## 3. Results

### 3.1. Temperature of the Structure

First, we simulate Structure I (Figure 2a). To analyze the heating of the structure and the time constants of the system, we consider a constant SOT current density of $1.1\cdot 10^{12}\;{\mathrm{Am}}^{-2}$ in the HM layer, in agreement with [8]. The potential at the top of the contact is left floating. After the current pulse is turned on, the temperature of the structure rises. Figure 3 shows an FL temperature increase ΔT in time. The black dots indicate data extracted from [8]. Only the average FL temperatures are shown. In order to match the heating curves, the conductivity of the surrounding oxide layer is varied. The solid blue and orange curves represent the heating of the structure with the surrounding oxide conductivity of 2.4 and 2.6 ${\mathrm{Wm}}^{-1}K^{-1}$, respectively. The data are fitted with a triple exponential (single and double exponentials do not provide a good match) and corresponding time constants $\tau_{i}$-s are extracted and listed in Table 1. The longest time constant $\tau_{3}$ is also plotted in Figure 3, for illustration.

We observe a relatively good match of $\tau_{3}$, whereas $\tau_{1}$ and $\tau_{2}$ show a bigger difference. We attribute the bigger difference for the fast time constants to the shape of the current pulse used (heavyside versus slower initial increase), although other differences are likely to be present due to parameter deviation from the original model (not listed in [8]). We note that the fast time constants mostly represent the heating of the structure, whereas a slow temperature increase is present due to a slow temperature increase of the surrounding oxide.

In order to fully understand the switching of the SOT-MRAM cell, we focus on the more complex Structure II, described in Section 2.2. We first apply only the SOT voltage $U_{\mathrm{SOT}}$ between the lower contact and the write line, which results in the SOT current $I_{\mathrm{SOT}}$ through the HM. In Figure 4a, the temperature of the structure at 0.2 ns for 0.4 V is illustrated. The temperature increases fast around the FL, mainly due to the heating in the HM. The sides of the HM are cooled down by heat transfer through the via and the current line. The heat transfer through the MgO layer is not significant and a strong temperature gradient across the layer exists due to its very low thermal conductivity. The FL temperature increase for different voltages is shown in Figure 4b. We observe a swift temperature increase in the beginning and a much slower increase towards the end. As the used model is linear, the temperature increase is proportional to the heating power in the system—proportional to the second power of $U_{\mathrm{SOT}}$, or $I_{\mathrm{SOT}}$. The linear dependence of the FL temperature increase ΔT with respect to $U_{\mathrm{SOT}}^{2}$ is shown in the inset of Figure 4b, similar to [46]. In other words, if only the $I_{\mathrm{SOT}}$ is considered throughout the structure, it is sufficient to scale the temperature with its second power. This result is, however, not applicable if both $I_{\mathrm{SOT}}$ and $I_{\mathrm{STT}}$ are present, as shown in the following subsections. We notice the time constants of Structure II (last line in Table 1) are shorter than those of Structure I, mainly due to the structure geometry and different boundary (Structure I unrealistically floats in the oxide). As previously mentioned, the two short time constants are dominated by the smaller HM and MTJ sizes, whereas the long time constant is mainly determined by the heating of the contacts, substrate, and the surrounding oxide.

### 3.2. Effect of Temperature on the Initial Switching Dynamics

In this section, we investigate how the increasing temperature affects the initial stage of the switching of Structure II (Figure 2b). First, we consider $I_{\mathrm{SOT}}$ only and no heating of the structure (no scaling applied). In Figure 5a, the initial dynamics for different $U_{\mathrm{SOT}}$ is shown. An average FL magnetization in the *z*-direction is shown.

A sharp transition into a final state can be observed with respect to $U_{\mathrm{SOT}}$. For the lower voltages, the FL magnetization starts to oscillate; however, it falls back close to the negative *z*-direction. When the $U_{\mathrm{SOT}}$ = 0.6 V is applied, the magnetization suddenly flips in-plane ($m_{z}=0$). In order to understand this behavior, we employ a simpler macromagnetic model. The steady-state solutions of the explicit form of the LLG are shown in Figure 6. The black dots represent a stable solution, whereas the gray dots show an unstable solution. The colored markers represent the final magnetization states of the system and the magnitude of the first oscillation (the strongest one) for different damping and different pulse shapes. The crosses show a solution when a heavyside function is applied. The final solution falls on the lower branch, until the first oscillation comes close to the unstable one. When the oscillation is big enough, the final magnetization solution falls into the plane ($m_{z}$ = 0)), thus causing the abrupt change in the final state of the system. When the damping is increased, the first oscillation is reduced and the jump is present for higher current densities. Lastly, if the voltage is changed slowly (1 ns for the full value), the initial oscillation is almost gone and the solution follows the lower stable solution, until it disappears for high enough current densities.

When the full temperature simulation is included, the magnetization dynamics change significantly (Figure 5b). The sharp in-plane transition is not present anymore. For higher voltages, the magnetization flips immediately into the plane whereas, for the lower voltages, an incubation phase can be observed, in agreement with experimental data [8]. We observe significantly lower voltages required for the magnetization in-plane flip in comparison to the constant temperature model. For the full temperature model, a voltage of 0.34 V is able to bring magnetization in-plane versus the 0.6 V required in the constant temperature model. If one considers the shorter time scales, the reduction is less pronounced; however, even the 0.42 V means a significant reduction in the critical SOT current. The lower critical switching voltage is caused by reduced anisotropy energy and saturation magnetization, which moves the shoulder of the unstable solution lower. The lowered exchange stiffens then enables easier nucleation of the magnetization reversal and allows for a more domain-like switching.

Lastly, we compare the reduction in the critical switching voltages to the results of the highly damped macrospin model [47]. Within the macrospin approximation, the critical switching current is proportional to $M_{S}$ and $K_{a}$ in the absence of the external field $H_{ext}$.
(14)$$J_{C}=\frac{2e\;M_{S}\;t_{F}}{\hbar \;\theta_{SHA}}\left(\frac{H_{aniso}}{2}-\frac{H_{ext}}{\sqrt{2}}\right)$$
where $t_{F}$ stands for the thickness of the FL. First, we consider the switching at 0.42 V when no significant initial oscillation is present. Considering the FL temperature reached when the FL starts switching (∼355 K), $M_{S}$ and $K_{a}$ drop to 91 and 76% of the initial value, respectively, and the critical switching current (voltage) is reduced to 69%. This is in good agreement with our simulation, with the critical voltage reduced to 0.42/0.6 = 70% of its initial value. Secondly, we look at the switching at 0.34 V. When the FL switches, the FL temperature reaches ∼345 K, which results in a critical voltage reduced to about 74% of its original value. The reduction is significantly lower than the simulated reduction of 0.34/0.6 = 56%. We attribute this discrepancy to the missing dynamics in the critical current calculation within the macrospin model. As we have already demonstrated, the initial oscillation can significantly reduce the critical current, and we conclude that any further reduction in the critical current is caused by the increase in the amplitude of the FL magnetization oscillation.

### 3.3. Field-Free Switching—Combined STT-SOT-MRAM

We now focus on the previously mentioned field-free switching, often referred to as a combined STT-SOT switching or SOT-assisted STT switching. In the first switching phase, both the SOT and STT currents are present, whereas in the later switching phase, the SOT current is turned off. We employ both of the pulses for the first 2 ns and investigate the difference between the constant temperature and full temperature models again. We vary $U_{\mathrm{SOT}}$, whereas $U_{STT}$ is kept at a constant 0.75 V.

In Figure 7, the switching simulations within the described SOT-STT switching scheme are shown. We observe that the behavior of both models looks very alike (constant temperature model shown in Figure 7a, full temperature model in Figure 7b). The initial oscillatory behavior is present in both systems. We attribute this to the additional STT torque, which acts on the FL only when nonzero magnetization components $m_{x}$ and $m_{y}$ are present. The STT torque is strongest when the FL magnetization is in-plane. An important difference between the two models is still the significantly reduced critical voltage, which flips the FL magnetization in the plane: 0.3 versus 0.5 V for the full 2 ns interval, or 0.4 versus 0.6 V if a shorter 0.5 ns SOT interval is considered. The reduced critical voltage can clearly be seen from Figure 8a, which shows the average magnetic parameter change in the FL for $U_{\mathrm{SOT}}$ = 0.31 V and $U_{\mathrm{STT}}$ = 0.75 V. The anisotropy energy density is significantly reduced and so is the energy barrier. In Figure 8b, the corresponding temperature increase in the FL is shown (in black). The maximum and minimum FL temperatures are also indicated (gray dotted and dashed lines, respectively). We observe a highly nontrivial temperature development due to the STT heating and changing resistance of the MTJ. At 2 ns, a fast temperature decrease is observed due to the $I_{\mathrm{SOT}}$ being turned off.

### 3.4. Switching with External Fields

In this last subsection, we focus on switching assisted by an external field. Such a field is of importance as it can represent different scenarios: the real external field, the stray fields induced by additional hard magnetic layers above [8] or below [7] the MTJ stack, or doping [10].

First, we simulate Structure II with a 50 mT external field in the $I_{\mathrm{SOT}}$ direction. We only apply the $U_{\mathrm{SOT}}$ voltage and vary its length. Unlike with SOT only, the magnetization does not stay in-plane ($m_{z}$ = 0), but due to the presence of the magnetic field, it precesses around the field, which brings the magnetization closer to the reversed orientation. This can be understood from the first cross-product term in the LLG (Equation 1). The numerical experiment is as follows: (i) In the beginning, the system is left to relax for 1 ns. (ii) After the relaxation, $U_{\mathrm{SOT}}$ is switched on for a short period of time. (iii) $U_{\mathrm{SOT}}$ is switched off again and the system relaxes to the final state. We study two different $U_{\mathrm{SOT}}$ pulse durations, 1.5 and 3.0 ns. The results of such an experiment are shown in Figure 9. For the shorter 1.5 ns pulse, voltages 0.32 V and below do not switch. For 0.32 V, a short incubation is observed, the magnetization passes the in-plane configuration but is brought back to the initial state after the pulse is turned off. For the longer 3.0 ns pulse, both 0.32 V and 0.30 V switch due to the longer pulse length. We note that for the constant temperature model, voltages up to 0.4 V do not show any switching, and the switching simulations are therefore not shown.

As a last experiment, we simulate SOT switching with an external field of 50 mT (identical direction to $I_{\mathrm{SOT}}$) and with additional heating due to the STT. Such a system was reported in [28]. We keep both the $U_{\mathrm{SOT}}$ and $U_{\mathrm{STT}}$ for 4 ns and let the magnetization relax afterward. We then check the final magnetization state and see if the switching has been completed, and then we determine the lowest $U_{\mathrm{SOT}}$ for which the magnetization is switched, the critical SOT switching voltage $U_{\mathrm{SOT}}$. We note that the temperature increase consists of both the smooth SOT temperature change shown in Figure 4b and the FL magnetization-dependent STT heating. The total temperature increase is similar to the one reported in Figure 8b but without the drop due to the SOT being turned off. In Figure 10, the dependence of $U_{\mathrm{SOT}}$ on $U_{\mathrm{STT}}$ is shown. We observe a parabolic dependence of $U_{\mathrm{SOT}}^{C}$ with the increasing absolute value of $U_{\mathrm{STT}}$, in agreement with [28]. Note that our system always kept the same set of parameters. In comparison to [28], where an additional voltage-controlled magnetic anisotropy (VCMA) was considered, we have a stronger STT torque, and no VCMA. The qualitatively identical result shows the significance of the additional STT heating on the $U_{\mathrm{SOT}}^{C}$. We observe a slight shift of the parabola due to the additional STT torques, in agreement with [28].

In all the previously presented experiments, the increased temperature, due to Joule heating, significantly reduces switching currents and/or switching times. In turn, the writing energy of the SOT-MRAM cell can be reduced if the pulse duration is kept short. In addition, the reduced switching currents and voltages allow for smaller switching transistors needed for memory operation, resulting in a reduced device footprint. On the other hand, all the generated heat is an irreversible energy loss that contributes to the total switching energy. Moreover, the increased temperatures can lead to a random bit flip just after the current pulse is turned off before the FL cools down, resulting in an increased write error rate. Finally, if the temperature increase is too high, the memory device can be damaged.

## 4. Conclusions

We coupled magnetization, charge, spin, and temperature dynamics to study modern SOT-MRAM devices. In order to allow for the SOT switching, the spin Hall effect was added. To account for the elevated temperature, scaling of the magnetization parameters was implemented. We validated our model by comparing the temperature dynamics to the previously published data and obtained a good agreement. With the validated model, we showed that a significant incubation time is present when the SOT-generating current is applied to the MRAM cell and that such behavior cannot be reproduced by a constant temperature model. We then demonstrated a field-free switching combining both SOT and STT. The temperature shows a nontrivial behavior due to the change in MTJ resistance during switching. Due to the increased temperature, the critical switching voltage is significantly reduced in comparison to the constant temperature model. We also studied SOT-MRAM switching with an external field. We changed the SOT voltage duration and showed that the increasing temperature has significant effects on the switching dynamics. Finally, we showed that an additional STT voltage pulse reduces the critical SOT switching voltage due to the extra heating. This dependence is quadratic, in agreement with our experiments. We conclude that Joule heating significantly affects the switching behavior of SOT-MRAM cells and can considerably reduce the switching currents and switching times, resulting in reduced writing energy and total device size. However, special care should still be taken regarding the maximum temperature to prevent any damage to the device. In order to fully understand the switching of modern SOT-MRAM devices and to accurately predict and optimize the intended behavior, the full temperature dynamics have to be considered.

## Figures and Tables

**Figure 1 micromachines-14-01581-f001:**
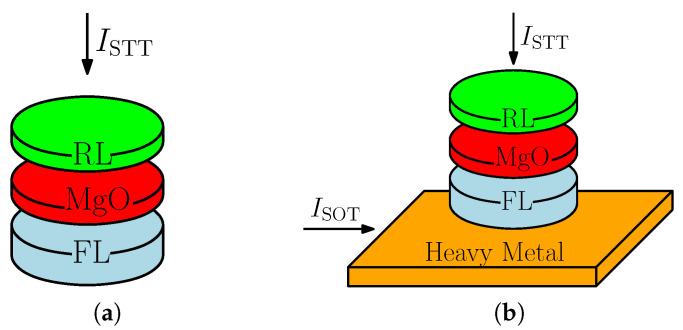
(**a**) A schematic illustration of STT- and (**b**) SOT-MRAM cells. Two separate current paths are present for the SOT-MRAM, where an HM layer is placed underneath the magnetic FL.

**Figure 2 micromachines-14-01581-f002:**
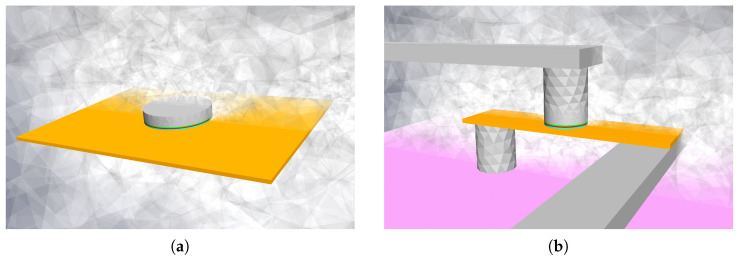
Simulated SOT-MRAM structures. (**a**) Simple structure from [8]. (**b**) Realistic structure with contacts, current lines, and a Si buffer beneath.

**Figure 3 micromachines-14-01581-f003:**
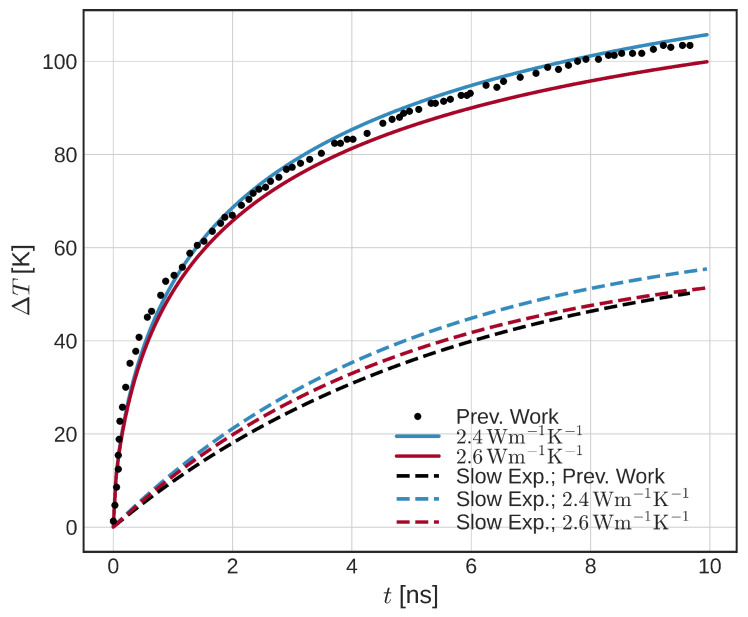
Temperature increase of Structure I [8]. Comparison between the data extracted from [8] (black dotted) and the presented model with two different conductivity of the surrounding oxide (solid). The slow exponential temperature increases are extracted (dashed). A constant current density of $1.1\times 10^{12}\;{\mathrm{Am}}^{-2}$ in the HM is considered.

**Figure 4 micromachines-14-01581-f004:**
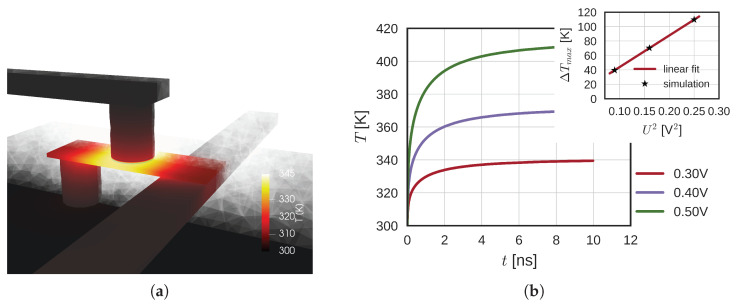
(**a**) Structure temperature at 0.2 ns after a voltage pulse $U_{\mathrm{SOT}}$ = 0.4 V was applied between the contacts. In the beginning, the temperature increase is centralized around the MTJ stack. (**b**) Maximum temperature increase of the FL for different voltages. The inset shows the maximum temperature increase with respect to $U_{\mathrm{SOT}}^{2}$.

**Figure 5 micromachines-14-01581-f005:**
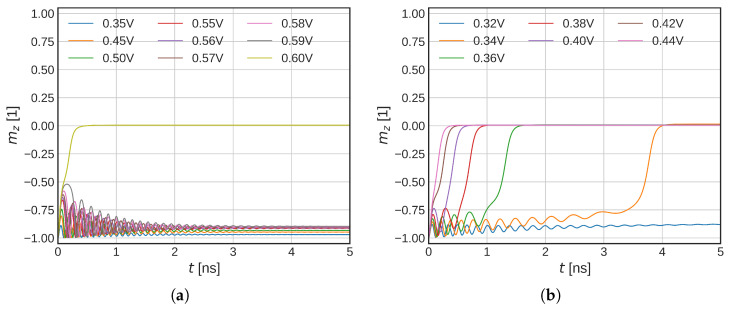
(**a**) Simulations of the FL magnetization in-plane flip with SOTs only. Temperature scaling is not included. (**b**) Simulations of the FL magnetization in-plane flip with SOTs only, with temperature scaling included. An incubation time due to the slow temperature rise can be observed. The critical SOT voltage that flips the FL magnetization in-plane is significantly reduced in comparison to the constant temperature model.

**Figure 6 micromachines-14-01581-f006:**
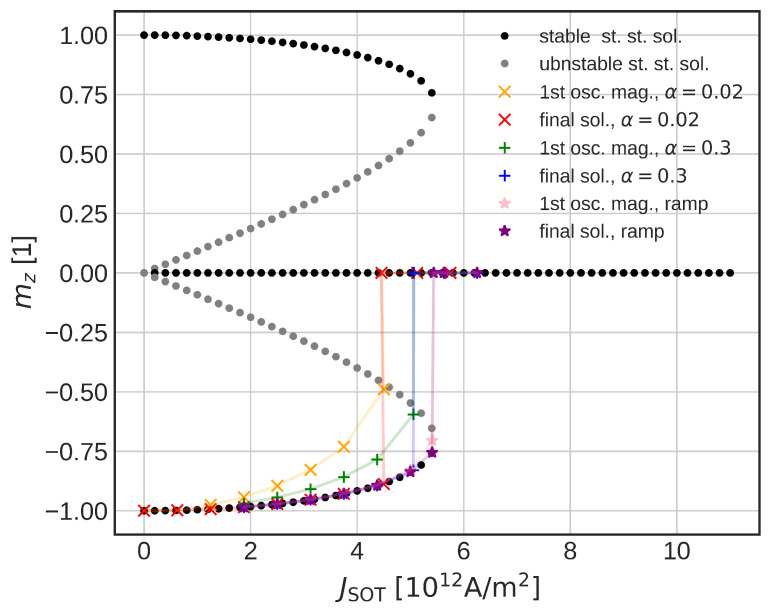
Steady-state solutions (gray, black) and real trajectories of the final magnetization state (orange, red, green). When the first oscillation amplitude (orange, green, pink) reaches the unstable solution (gray), an instant jump into the in-plane state appears.

**Figure 7 micromachines-14-01581-f007:**
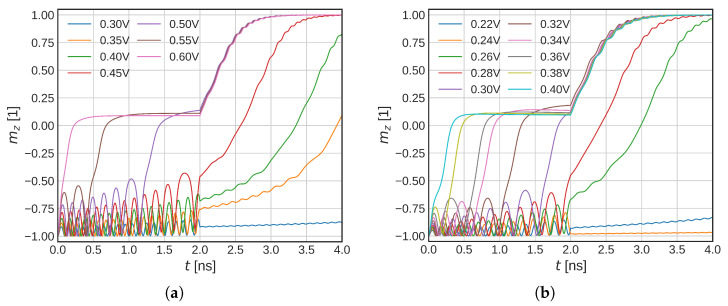
(**a**) Simulations of the combined STT-SOT switching at constant 300 K, and (**b**) with the full temperature model. The different paths (colors) represent different $U_{\mathrm{SOT}}$, $U_{\mathrm{STT}}$ = 0.75 V. The SOT current is only present during the first 2 ns, while the STT is kept for the whole simulation. Due to the additional presence of the STT field, the initial oscillatory behavior is amplified and acts as an additional field that amplifies the oscillations. Both of the switching simulations with a constant temperature and with the full temperature model look similar; however, the oscillations are modulated for the latter. The critical switching voltage is also significantly reduced.

**Figure 8 micromachines-14-01581-f008:**
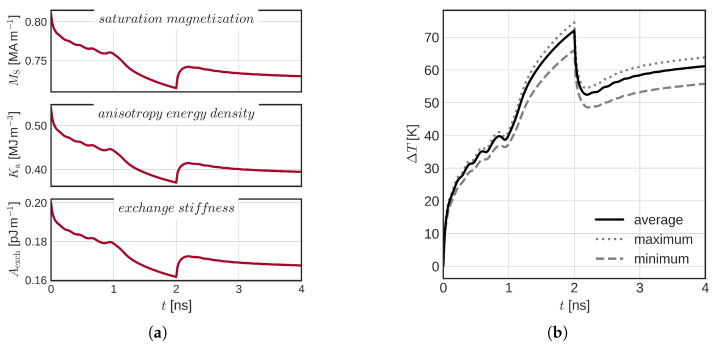
(**a**) Parameter change due to the increased FL temperature, and (**b**) the corresponding FL temperature increase.

**Figure 9 micromachines-14-01581-f009:**
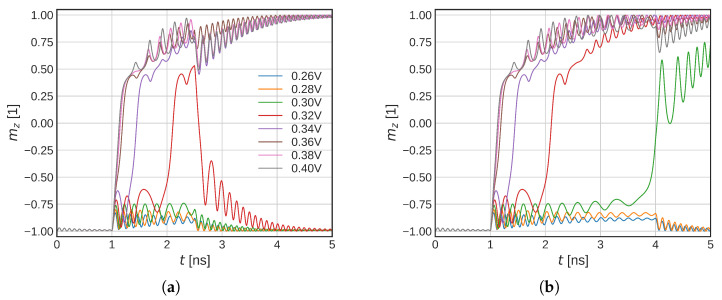
Switching simulations with an external field for different $U_{\mathrm{SOT}}$ and (**a**) 1.5 ns, (**b**) 3.0 ns pulse durations. The external field is 50 mT in the $I_{\mathrm{SOT}}$ direction. We let the system relax for 1 ns before the pulse is applied. The color coding is identical for both plots.

**Figure 10 micromachines-14-01581-f010:**
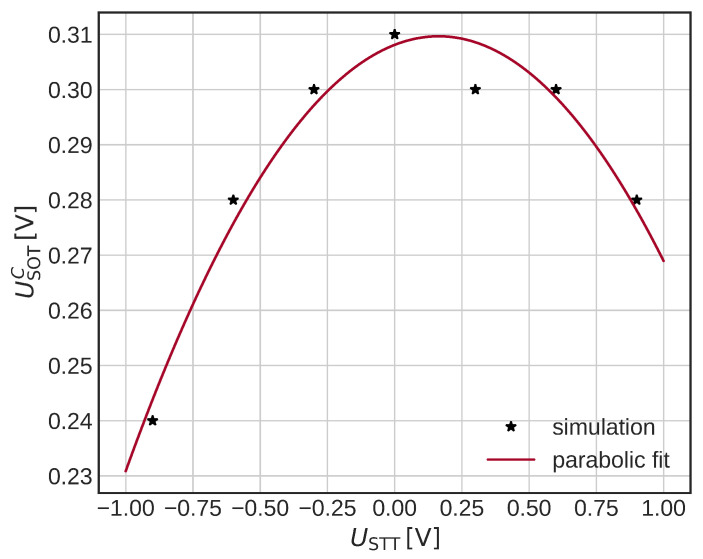
Dependence of the critical SOT switching voltage $U_{SOT}^{C}$ on the applied STT heating voltage for the SOT switching with an external field. A parabolic reduction in $U_{SOT}^{C}$ is observed due to the increased temperature of the FL. The parabola is shifted due to the additional STT torque.

**Table 1 micromachines-14-01581-t001:** Time constants for different structures.

Structure	$\tau_{1}$ (ns)	$\tau_{2}$ (ns)	$\tau_{3}$ (ns)
Structure I, 2.4 ${\mathrm{Wm}}^{-1}K^{-1}$	0.073	0.746	5.013
Structure I, 2.6 ${\mathrm{Wm}}^{-1}K^{-1}$	0.072	0.733	4.896
Structure I, [8]	0.152	1.216	5.796
Structure II	0.035	0.439	2.539

## Data Availability

The datasets generated and/or analyzed during the current study are available from the corresponding author upon reasonable request.

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
