# Peer review of "A Comprehensive Study of Temperature and Its Effects in SOT-MRAM Devices"

_micromachines, 2023, doi:10.3390/mi14081581_

Round 1
Reviewer 1 Report
In this manuscript, the authors reported the modeling of temperature and its effects on field-free and field-assisted switching of SOT-MRAM by employing the three-dimensional model coupling magnetization, charge, spin, and temperature dynamics. The motivation of this work is well articulated in the introduction, followed by a detailed method description. In the result part, they give comprehensive analyzes of their simulated results. The authors validated their model by comparing the temperature dynamics to the previously published data and obtained a good ag reem with experimental results reported in Ref. 8. Moreover, they demonstrated the phenomenon that the critical SOT switching voltage is significantly reduced by the increased temperature for switching of field-free SOT-MRAM and that the critical SOT voltage shows a parabolic decrease with respect to the voltage applied across the magnetic tunnel junction (MTJ) of the SOT-MRAM cell with an external-field assisted switching. The key conclusion is that full temperature dynamics have to be considered in order to fully understand the switching of modern SOT-MRAM devices and accurately predict the correct behavior. This work contributes to fully understanding the switching of the SOT-MRAM cell. The whole manuscript is also well organized. In my opinion, it can potentially be published in Micromachines after addressing the following comments.
Considering the important role of temperature on the switching of SOT-MRAM mentioned in this manuscript, can the author give a discussion about the advantages and disadvantages of the Joule heating effect on the SOT-MRAM switching?
Besides, in order to enhance the thermal stability of the SOT devices, there are many research works about field-free switching in perpendicular single layers very recently. [Nat. Commun. 12, 4555, 2021; Nat. Commun. 13 (1), 3539, 2022; Adv. Funct. Mater. 32, 2200660, 2022; ACS Nano 17 (7), 6400-6409, 2023]. Some single layers exhibit high bulk PMA, and the industry-required thermal stability can be achieved even when the memory element is smaller than 20 nm in diameter. The authors may want to review these works in the introduction.
Moreover, for the reader's convenience, the authors may want to give a detailed literature review and citation. For example, in the "a combination of STT and SOT switching, or combinations of multiple methods" the original articles should be cited also.
Author Response
Please, find the responses in the attached document.

Reviewer 2 Report
The original work by T. Hadámek et al presents numerical simulations about the effect of temperature in SOT-RAM. The results are well presented, and the paper is organized making easy to follow the main points risen by the authors. They are transparent and provide the necessary tools to repeat their work.
Thermal effects in SOT-MRAMs have been widely studied, in both experimental and theoretical works, as it is shown in S.A. Razavi et al Phys. Rev. App. 7, 024023 (2017), S.K. Ziaur Rahaman et al IEEE Electron Device Letters 39, 9 (2018), or T.H. Pham et al, Phys. Rev. App. 9, 064032 (2018); and even in antiferromagnetic materials S. Arpaci et al., Nat. Comm 12, 3828 (2021). Then, I think their work lacks new results and just confirms something the spintronics community have known and taken into consideration fabricating devices during the last decade. The only interest I see for their paper is the computational tool developed by the authors, what can be of interest for design proposes.
For this reason, I cannot recommend the paper for its publication.
The English of the manuscript is adequate.
Author Response

(The authors gave the same response as above.)

Reviewer 3 Report
The manuscript reports on a significant incubation time to the switching of SOT-MRAM which is introduced by the relatively slow temperature increase after a rapid initial temperature rise. The manuscript also reports that comparing with the constant temperature model, the critical switching voltage of field-free SOT-MRAM is decreased with the increased temperature. Furthermore, the SOT-MRAM switching with an external field is also discussed. The critical SOT switching voltage reduces with the STT voltage pulse because of the external heating. However, there are still a few things should be improved. My concerns, questions and suggestions are shown below.
1. The methods of breaking symmetry mentioned in the introduction are not comprehensive. The following methods are also suggested to be cited in this article, including exchange bias (Nat. Mater. 15, 535-541, (2016)), interlayer exchange coupling (Nat. Nanotechnol. 11, 758-762, (2016)), spin current gradient (Nat. Mater. 16, 712 (2017)), lateral spin–orbit torques (Adv. Mater. 32, e1907929, (2020)), competing spin currents (Phys. Rev. Lett. 120, 117703 (2018)), out-of-plane spin polarization (Nat. Mater. 17, 509-513, (2018); Nat. Nanotechnol. 16, 277-282, (2021); Nat. Mater. 21, 1029-1034, (2022)), and ion implantation (Phys. Rev. Applied 15, 054013 (2021), Appl. Phys. Lett. 120, 062402 (2022)).
2. In the section 3.2, the author mentioned that “the lower critical switching voltage is mainly caused by the reduced anisotropy energy and saturation magnetization”. However, any supporting results in the simulations? Provide additional information for the speculation, please.
3. For the additional heating due to the STT which is mentioned in section 3.4, how to quantify the temperature which is introduced by the STT? And the SOT voltage can also introduce the increased temperature of the write line and FL, there should be more information to explain the contribution of SOT and STT on the additional heating of MTJ.
In summary, this is an interesting study, but the presentation of this manuscript requires improvements and also supportive information.
This is an interesting study, but the presentation of this manuscript requires improvements and also supportive information.
Author Response

(The authors gave the same response as above.)
